# Identification of Risk Factors to Predict the Occurrences of Relapses in Individuals with Schizophrenia Spectrum Disorder in Iran

**DOI:** 10.3390/ijerph18020546

**Published:** 2021-01-11

**Authors:** Omran Davarinejad, Tahereh Mohammadi Majd, Farzaneh Golmohammadi, Payam Mohammadi, Farnaz Radmehr, Mostafa Alikhani, Tayebeh Motaei, Mehdi Moradinazar, Annette Brühl, Dena Sadeghi Bahmani, Serge Brand

**Affiliations:** 1Substance Abuse Prevention Research Center, Health Institute, Kermanshah University of Medical Sciences, Kermanshah 6719851115, Iran; odavarinejad@gmail.com (O.D.); m.alikhani18@yahoo.com (M.A.); 2Clinical Research Development Center, Imam Khomeini and Mohammad Kermanshahi and Farabi Hospitals, Kermanshah University of Medical Sciences, Kermanshah 6719851451, Iran; tahereh_mohammadi@yahoo.com (T.M.M.); f.golmohammadi1367@gmail.com (F.G.); Payammohamadi1142@gmail.com (P.M.); radmehr.f12@gmail.com (F.R.); Tmotaee2020@gmail.com (T.M.); m.moradinazar@gmail.com (M.M.); 3Center for Affective, Stress and Sleep Disorders (ZASS), University of Basel, 4002 Basel, Switzerland; annette.bruehl@upk.ch (A.B.); dena.sadeghibahmani@upk.ch (D.S.B.); 4Sleep Disorders Research Center, Kermanshah University of Medical Sciences, Kermanshah 6719851115, Iran; 5Departments of Physical Therapy, University of Alabama at Birmingham, Birmingham, AL 35209, USA; 6Department of Sport, Exercise, and Health, University of Basel, 4002 Basel, Switzerland; 7School of Medicine, Tehran University of Medical Sciences, Tehran 1416753955, Iran

**Keywords:** schizophrenia spectrum disorder, relapse, predictions, treatment adherence

## Abstract

Schizophrenia Spectrum Disorder (SSD) is a chronic psychiatric disorder with a modest treatment outcome. In addition, relapses are commonplace. Here, we sought to identify factors that predict relapse latency and frequency. To this end, we retrospectively analyzed data for individuals with SSD. Medical records of 401 individuals with SSD were analyzed (mean age: 25.51 years; 63.6% males) covering a five-year period. Univariate and multivariate Penalized Likelihood Models with Shared Log-Normal Frailty were used to determine the correlation between discharge time and relapse and to identify risk factors. A total of 683 relapses were observed in males, and 422 relapses in females. The Relapse Hazard Ratio (RHR) decreased with age (RHR = 0.99, CI: (0.98–0.998)) and with participants’ adherence to pharmacological treatment (HR = 0.71, CI: 0.58–0.86). In contrast, RHR increased with a history of suicide attempts (HR = 1.32, CI: 1.09–1.60), and a gradual compared to a sudden onset of disease (HR = 1.45, CI: 1.02–2.05). Gender was not predictive. Data indicate that preventive and therapeutic interventions may be particularly important for individuals who are younger at disease onset, have a history of suicide attempts, have experienced a gradual onset of disease, and have difficulties adhering to medication.

## 1. Introduction

Even 100 years after Kreapelin’s seminal work on the phenomenon of *dementia praecox*, schizophrenia (schizophrenia spectrum disorder; SSD) remains enigmatic in regards to its multiform etiology, its symptomatology, its uncertain disease progress, the modest treatment outcomes, and the issue of comorbidities such as tobacco use disorder, obesity, and diabetes [1,2,3]. The term schizophrenia spectrum disorder covers a broad range of different symptoms and not all individuals with SSD display all the symptoms and not all symptoms are simultaneously present. Typically, individuals with SSD show disorganization in formal thoughts and language, hallucinations, delusions, catatonic symptoms, dysfunctions in affect and mood, self-disorder, somatic symptoms, and neurocognitive impairments. Up to 50% of individuals with SSD display functional impairments that increase the risk of permanent unemployment and inability to build and maintain stable relationships [2,3].

Olabi et al. [4] further stressed that SSD is not a uniform disorder and that interindividual variance of the disorder was also mirrored in the high variance of results among studies.

Despite progress in describing the brain morphology of individuals with SSD, distinctive features identified in imaging studies do not appear to be associated with either the incidence or prevalence of relapses [3,5,6]. On this point, in their exploratory quantitative meta-analysis of structural neuroimaging studies, Chee et al. [7] concluded that neuroanatomical indices that are predictive of relapses in individuals with SSS have yet to be identified. It follows that the prediction of relapses and the time interval between recovery and relapse cannot rely on imaging studies, but must do so on observational evidence. Eisner et al. [8] reported that relapse in psychosis is a common phenomenon among individuals with SSD. Furthermore, relapses are associated with higher risks of unemployment, long-term deterioration, and suicidal behavior. Given this, detecting early signs and predictors of a future relapse deserve careful attention, and this holds particularly true for Iran, where research on this topic is virtually nonexistent.

Leucht et al. [3] identified the following common early signs of relapse: Restlessness, sleep disorders, tension, issues at work, feelings of not being understood and being stressed, low degree of satisfaction, social withdrawal, cognitive impairments such as in attention and memory, increased religious thoughts, losing control of oneself, hallucinations, external voices, and thoughts taking control. Emsley et al. [9] concluded from their narrative review that the risk of relapses was higher if medication was interrupted, irrespective of the number of previous psychotic episodes. In addition, and against expectations, a longer treatment period prior to medication interruption did not reduce the odds of relapse; relapses may occur shortly after discontinuing medication with either no or several warning signs. In 1995, Jablensky [10] reviewed the state-of-the-art and identified six factors associated with positive and negative outcomes: Odds of relapse were greater when the individual with SSD was single, divorced, male, and living in a family environment with high expressed emotion (factor: Sociodemographic and family-related data), when social withdrawal was high and adjustment problems were observable in adolescence (factor: Premorbid personality and adjustment problems), when relapses were more frequent and longer (factor: Preceding illness characteristics), when the onset of disease was either very gradual or abrupt (factor: Onset characteristics), when negative symptoms and acoustic hallucinations prevailed (factor: Initial clinical frame), and when cannabis use was reported (factor: Other characteristics). To conclude, Jablensky’s overview summarized a broad range of possible factors consistent with what had been already observed in the mid-70s [11]. A total of 47 different predictors were identified and these predictors explained 38% of the variance in relapses, which leaves 62% unexplained. The same report also notes that 60% of inpatients with SSD suffered from a relapse after their first hospitalization.

Lecomte et al. [12] conducted a metareview of 31 meta-analyses summarizing the results from 3044 papers on relapse predictors. For protective factors, they concluded there was moderate to strong evidence for antipsychotic medication in adults, family interventions, social skills training, and interventions focusing on recovery management skills. Eisner et al. [8] conducted in-depth interviews with individuals with SSD to detect, beyond indicators related to sociodemographic factors or medication, possible early cognitive, emotional, or behavioral signs of an imminent relapse. Their interviews identified the following signs: Increased indecisiveness about insignificant choices; poor multitasking; thought interference; disturbance of receptive speech; increased stress reactivity; hypersensitivity to sounds; visual problems such as straight things appearing crooked and double vision; thought perseveration; overly distracted by stimuli; disturbances of olfactory, gustatory, and tactile perceptions; micropsia and macropsia; near- or tele-vision; shapes appearing different or distorted; decreased ability to distinguish between ideas and perceptions or between fantasy and true memories; derealization; thought blockages and thought pressure; and slowed-down thinking.

Others [3,10,13,14] have reported that the likelihood of suffering from SSD is greater when first to third degree relatives have been diagnosed with SSD.

Mortensen et al. [13] showed that the odds of suffering from SSD are greater when a person grows up in an urban area (place of birth effect), and when a person was born in the spring but not in the autumn (season effect).

Lastly, David and Prince [15] concluded in their review that the association between head traumas and the occurrence of SSD is weak and that results are contradictory.

Currently there is some but modest and inconsistent evidence that sociodemographic factors (e.g., younger age, male gender, living in an urban area, and being single, born in the spring), illness-related factors (e.g., slow and prolonged prodromal phase, non-adherence to medication, history of SSD in first to third degree relatives), both typical (thought interference, increased stress reactivity, thought perseveration; see Eisner et al. [8]) and atypical symptoms (e.g., social withdrawal, symptoms of depression, issues at workplace) can all increase the odds of a relapse. We took these observations into consideration and introduced age, sex, marital status, mode of disease onset, medication adherence, place and season of birth, history of SSD among relatives, history of head traumas, and suicide attempts as possible factors affecting the odds and latency of relapses.

In regards to the situation in Iran, the prevalence of SDD ranges from 05% to 0.6% to 0.89%, depending on the diagnostic criteria (lower prevalence rates based on the DSM-IV than on the DSM-5), and the date of the survey (more recent studies report higher prevalence rates [16]). Mohammadi et al. [17] assessed 25,180 Iranian adults via face-to-face clinical interviews and reported a prevalence rate of 0.89% for SSD.

Furthermore, in regards to research on relapse prediction in Iran, to our knowledge, only Rahmati et al. [14] have considered this. This might not be surprising, given the low prevalence rate of SDD compared to the prevalence rates for anxiety disorders (8.35%), mood disorders (4.29%), and psychiatric disorders in general (10.81%) [17]. However, given a population of about 86 million, statistically 765,400 individuals might be suffering from SSD. Rahmati et al. [14] analyzed medical records of 159 individuals with SSD (mean age: 21.52 years; age range: 10 to 43 years) over a period of about six years. The best predictors for a first relapse were a younger age at disease onset, being male, being single, a gradual and prolonged prodromal phase, and a family member with a history with SSD. Two and more relapses were predicted by these factors and by the occurrence of a preceding relapse.

Taken together, in regards to the current situation for SSD and the risk of relapses in Iran, research is particularly scarce. To counter this, the present study reviewed retrospectively the medical records of 401 individual with SSD over a period of about 60 months in order to identify factors predicting a relapse and the latency of relapses.

Based on previous studies [9,12,14] our first hypothesis was that a younger age at disease onset, low adherence to prescribed medication, being male, and a gradual and prolonged prodromal disease phase would predict the odds and number of relapses. Following Eisner et al. [8], our second hypothesis was that the occurrence of suicide attempts would predict the odds and number of relapses.

We believe that the findings from this study could be of clinical and practical importance because identifying robust predictors of relapses should allow for a more effective and timely prevention.

## 2. Method

### 2.1. Procedure

This is a retrospective cohort study based on medical records of individuals with SSD who were admitted at least once to the Farabi Hospital of Kermanshah (Kermanshah, Iran) between 2015 and 2019. We analyzed anonymized medical records of all individuals with SSD, who were readmitted at least once. Participants signed a written informed consent and the Research Department of the Kermanshah University of Medical Sciences (KUMS; Kermanshah, Iran) approved the present study (registration code: IR.KUMS.REC.1398.329), which was performed in accordance with the seventh and current revision [18] of the Declaration of Helsinki.

### 2.2. Data Collection Based on the Medical Records: Variables of Interest

The following information was extracted from the medical records both for the first and all consecutive hospitalizations.

Sociodemographic information: Age; sex; civil status (single; married; divorced/widowed); place of birth (rural vs. urban area); season of birth (spring; summer; autumn; winter); and employment status (employed vs. unemployed).

SSD-related and illness-related information: Mode of disease onset (gradual vs. acute onset); adherence to prescribed medication (yes vs. no; if medications were either antipsychotics, anticonvulsants, or a combination of these); substance abuse as secondary comorbidity (yes vs. no); number of relapse(s); time lapse(s) between the current re-hospitalization and the last discharge; suicide attempt(s) (yes vs. no); family history of SSD (first to third degree relatives: yes vs. no); and history of head trauma (skull fracture before disease onset or coma after the trauma: yes vs. no).

### 2.3. Definition of Relapse

Rehospitalization was used as a proxy for relapse and we were interested in the number of such relapses and the timing between hospitalization admission. We followed Rahmati et al. [14] in defining a relapse as the first SSD-related re-hospitalization of an individual with SSD after discharge. Outcome measures were the number of relapses and the time interval between initial discharge and relapse(s).

### 2.4. Definition of Medication Adherence

Medication non-adherence was defined as the gap between the patient’s reported medication intake and the prescribed medication intake [19,20].

### 2.5. Sample Selection

Between 2014 and 2019, 902 records of inpatients with SSD were stored in the hospital archive. The inclusion criteria were: 1. Diagnosis of SSD, as ascertained by a trained and experienced psychiatrist or clinical psychologist and based on the DSM 5 [21]. 2. Age between 18 and 65 years. 3. General signed written informed consent. 4. At least one readmission on record. The exclusion criteria were: 1. Wrong diagnosis. 2. Unable or unwilling to answer some questions asked either on the phone or face-to-face in order to fill gaps in information missing from the medical records. 3. No readmission. As shown in Figure 1, of the medical records of 902 inpatients with SSD, 401 records (44.46%) were analyzed and 501 (55.54%) were excluded.

### 2.6. Statistical Analysis

For quantitative variables, we used means and standard deviations, while qualitative variables were used in order to analyze frequency and percentage. The variables were first univariate and then entered in a multivariate analysis in which the variables with *p*-value < 0.05 were introduced into the multivariate survival analysis. For data analysis, the recurrent event model was used employing Penalized Likelihood in software R.3.5.3 (R Foundation for Statistical Computing, Vienna, Austria) and “frailty pack”. Shared Log-Normal Frailty was used to analyze elapsed time to recurrent relapses. In this study, the time from onset of disease to onset of recurrent readmissions was recorded for each patient in days. The significance level was set at 0.05 for studying Adjusted Hazard Ratio.

## 3. Results

### 3.1. General Pbservations

Table 1 reports the descriptive statistical indices of sociodemographic and illness-related information, both for the entire sample and separately for gender.

Data of 401 individuals with SSD were analyzed and 255 (63.6%) were males (mean age: 37.56 years) and 146 (36.4%9) were females (mean age: 38.33 years). The numbers of male and female readmissions were 683 and 422 cases, respectively, as shown in Table 1.

### 3.2. Relapses

The median time to readmission was 238 days for individuals who were medication adherent, while for those with medication non-adherence readmission time was 106 days (*p* < 0.001; see Table 1). Almost all individuals with SSD had readmissions within a year after discharge (Figure 2).

The non-zero parameter in the Shared Log-Normal Frailty model (*p* < 0.001, Sigma Square = 0.08) indicated the effect of unknown individual factors and a significant correlation between the relapse time of SSD in the disease process. The model that does not consider the frailty of patients is not an appropriate model (Table 2).

With a greater age of disease onset, the Hazard Ratio of the disease relapse reduced, which was only 0.01 times (HR = 0.99, 95%, CI: 0.981–0.998).

Compared to a sudden onset, gradual onset increased the Hazard Ratio (HR = 1.45, 95%, CI = 1.02–2.05). A history of suicide attempts had a 1.32-fold higher risk of readmission to hospital (HR = 1.32, 95%, CI = 1.09–1.60).

The risk of subsequent relapses was 0.71-fold less for individuals with SSD with medication adherence than for those with no medication adherence (HR = 0.71, 95%, CI = 0.58–0.86). On the other hand, gender, marital status, employment status, substance abuse, head trauma, family history of schizophrenia, and season of birth had no effect on the Hazard Ratio of relapse (Table 2).

## 4. Discussion

The key findings of the present study were that over a period of 60 months, relapses were very common among individuals with SSD and that low adherence to medication, reported suicide attempts, a gradual onset of disease, and a younger age at disease onset all predicted a higher relapse ratio and a shorter relapse interval. In contrast, gender, marital status, employment status, substance abuse, head trauma, family history of schizophrenia, and season of birth did not predict the odds of relapses and/or relapse latency. The present results add to the current literature in an important way in that both treatment medication adherence and lack of suicide attempts were important factors in lower relapse rates and longer relapse intervals. While the pattern of results might be discouraging, we also note that it matches with what we know so far from statistical models predicting relapses in psychosis. Thus, Sullivan et al. [5] concluded in their systematic review that due to the lack of high quality evidence, it was impossible to formulate recommendations in regards to possible predictors. Similarly, it appears that evidence from neuroimaging has not generated robust predictors [6].

Two hypotheses were formulated and each of these is considered in turn.

Our first hypothesis was that being at a younger age at disease onset, low adherence to prescribed medication, male gender, and a gradual and prolonged prodromal phase of disease would predict the odds and number of relapses. This hypothesis was not, however, fully supported. While a lack of adherence to medication, younger age at disease onset, and a gradual onset of disease did all predict a shorter time interval between relapses, an independent effect of gender did not reach statistical significance. In this respect the present results are at odds with previous findings [9,12,14]. The evidence available to us from this study is insufficient to clarify why gender was not a predictor for relapses. However, Table 1 shows that while the gender ratio was 2(m):1(f) and thus very similar to the ratios reported in previous studies, in the present sample, descriptively, the females were more often divorced or widowed, more often unemployed, and more often reported a gradual onset of disease. In contrast, no descriptive differences in gender distributions were observed for the number of relapses or for medication adherence. Overall, the statistical analysis suggests that either there was in fact no gender difference or that other factors had a higher statistical power.

Medication adherence emerged in this study as a predictor of fewer relapses and longer inter-relapse intervals. Adherence to medication treatment is a therapeutic issue. The scientific community proposes two interventions (or their combination) for improving adherence to medication: Psychoeducation and long acting injectables (LAIs as opposed to oral antipsychotics (OAPs)). With respect to psychoeducation, Barkof et al. [22] concluded in their review that interventions with more sessions, sessions with a focus on medication adherence, and pragmatic interventions that focus on attention and memory problems improved adherence, while motivational interviewing did not appear to increase adherence. Similarly, Zhao et al. [23] showed that brief psychoeducational sessions (10 or less sessions) appeared to reduce the relapse rate in the medium term and to increase the medication compliance in the short term. Sendt et al. [24] showed in their systematic review that a patient’s positive attitude to medication and illness insight were associated with higher medication adherence. Among younger patients, a therapeutic relationship and social support had positive impacts on adherence to prescribed medication. In contrast, Hegerdüs and Kozel [25] found no association between adherence to therapy and adherence to medication. Similarly, Gray et al. [26] showed that adherence training as an adjunctive treatment reduced patients’ symptom severity but did not alter their attitudes to medication adherence. Lastly, Petry et al. [27] showed that financial inducements (USD 50 or more per week) for a longer duration yielded higher medication treatment adherence. In regards to LAIs, results from systematic reviews and meta-analyses showed that LAIs were superior to OAPs regarding hospitalization rate and all-cause discontinuation of medication but were not superior regarding the risk of hospitalization, time in hospital [28], or number of relapses [29] in RCTs, which by definition are less representative of real-world patients in naturalistic studies. Compared to second-generation OAPs, LAIs had lower relapse rates, a longer time to relapse, and fewer hospital days, but also a higher occurrence of extrapyramidal symptoms and prolactin-related symptoms [30]. In this context, second generation LAIs had similar effects to first generation LAIs, but the former dramatically increased serum prolactin concentrations, weight, and BMI (Body Mass Index) [31]. When compared to placebo, LAIs showed advantages with respect to psychosocial functioning, but this was not the case when compared to OAPs [32]. Compared to OAPs, LAIs reduced the percentage but not the absolute number of hospitalizations [33], while LAIs increased symptoms of dyskinesia [34].

Taken together, previous studies and present data showed that medication adherence appears to be an important protective factor for relapses, though when using second generation LAIs in particular the dramatic side effects such as weight gain, increased prolactin concentrations, and tardive dyskinesia should be carefully considered.

Our second hypothesis was that suicide attempts would predict the occurrence of relapses and shorter relapse intervals, and these expectations were confirmed. Thus, the present findings are in accord with previous results [8].

We believe that suicide attempts as a relapse risk demands particular attention. By definition, suicide is the decision deliberately to end one’s life. Suicide has become one of the leading causes of death in the world [35]. Furthermore, suicide attempts may have long-lasting negative consequences both for the individuals concerned and for those in their social worlds, including family members, workplace colleagues, and witnesses to the suicide attempt [35]. Importantly, when an individual has strong suicidal ideations, death appears to be the only solution to a subjectively perceived unbearable life situation [36].

Several factors have been advanced to predict suicide attempts (see [37,38] for extensive discussions). Here, we emphasize social and behavioral dimensions. Nazarzadeh et al. [39] identified loneliness and weak family ties as stronger predictors of suicidal behavior than co-varying economic factors. Early maladaptive schemas of emotional deprivation, social isolation, shame, and abandonment were related to a history of suicide attempts among patients with major depressive disorders [40]. Among women, reporting failed self-immolation, feelings of being trapped in the husband’s adverse family system, stress related to marital life, copycatting, and absence of social support were identified as key factors [41].

In addition, attempting and committing suicide implies behavior-inducing cognitions such as emotional stoicism, sensation-seeking, pain tolerance, and lack of fear of death (Deshpande, et al. [42]). Other social and behavioral factors include a state of over-arousal, along with social withdrawal and perceived burdensomeness [38], thwarted belongingness (i.e., the belief that one does not belong to a social group, or believes oneself to be unimportant and useless to other people) [43], and feelings of defeat, social rejection, entrapment, humiliation, and subjectively perceived low social support [44]. In this context, and in regards to the situation in Iran, individuals with SSD and their caregivers are frequently stigmatized and can often experience social rejection, covert and open hostility and aggression, and humiliation [45,46]. Given this, individuals with SSD and their caregivers and families risk further stigmatization, social isolation, and less access to health care [47].

We do not have from the present study the evidence to specify the psychological or physiological mechanisms that may underlie suicidal behavior. However, counseling and treatment of persons with SDD should pay particular attention to suicidal ideation, in particular to the patient’s quality and quantity of social interactions. This is especially important given the failure thus far of evidence from imaging studies to provide satisfactory predictors of either relapses or suicidal behavior [5,6]. Similarly, although 47 different predictors together explain about 38% of the variance in relapses and thus 62% of the variance remains unexplained [11], future studies should seek to identify other robust predictors of relapse and the means to improve the social competences to form and maintain stable social relationships of people with SSD.

The novelty of the study results should be balanced against the following limitations. First, rehospitalization is usually precipitated by an increase in positive symptoms of psychosis while other important domains of behavior and functioning such as cognitive performance, social interactions with family members, co-workers, neighbors, and health care providers, along with the health care system and the quality of care received after discharge from hospital, were not assessed. Likewise, secondly, the evidence available from the study cannot provide a robust and comprehensive answer to the question of whether SSD should be considered a neurodegenerative disease or instead a failed neuronal regenerative disease with the possibility of neuronal repair and regeneration or both [2]. Thirdly, given our statistical approach, only the records of participants with at least one rehospitalization were entered in the equations. Given this, it is possible that the pattern of results may be biased or imprecise. In this regard, fourthly, the pattern of results may be biased in two ways: (a) By definition a retrospective cohort study relies on the quality of medical records and (b) medical records and other SSD-related information are not available in cases where rehospitalization did not occur in the Farabi Hospital of Kermanshah (Kermanshah, Iran), but in another psychiatric hospital. Fifthly, the study did not address an additional but key question from the point-of-view of individuals with SSD and their caregivers: How to improve the quality of life of people with SSD. Besides the strategies for improving treatment adherence discussed above, regular physical activity appears to have a beneficial effect on aspects of SSD and in particular on unpleasant side-effects of antipsychotics such as weight gain and sedentary behavior [48,49,50,51,52,53,54,55]. Furthermore, there is a need for early detection and treatment of SSD symptoms so as to keep the quality of life in individuals with SSD as stable as possible [56].

## 5. Conclusions

Among a large sample of Iranians with SDD, younger age at disease onset, non-adherence to medication, a history of suicide attempts, and a gradual onset of disease all predicted higher hospital readmission and a shorter readmission interval after the initial admission. Given this, counseling and treatment should focus on patients distinguished by these risks. Only non-adherence can be addressed by preventive measures but it is hoped that these results will aid clinicians in their quest to improve treatment adherence.

## Figures and Tables

**Figure 1 ijerph-18-00546-f001:**
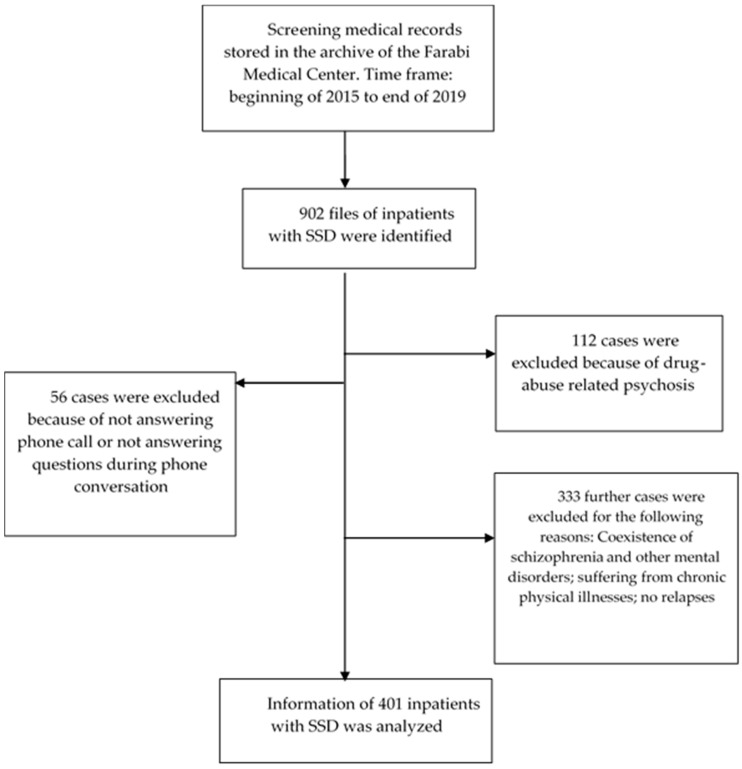
CONSORT diagram, data pool, and inclusion and exclusion criteria. (Schizophrenia Spectrum Disorder (SSD)).

**Figure 2 ijerph-18-00546-f002:**
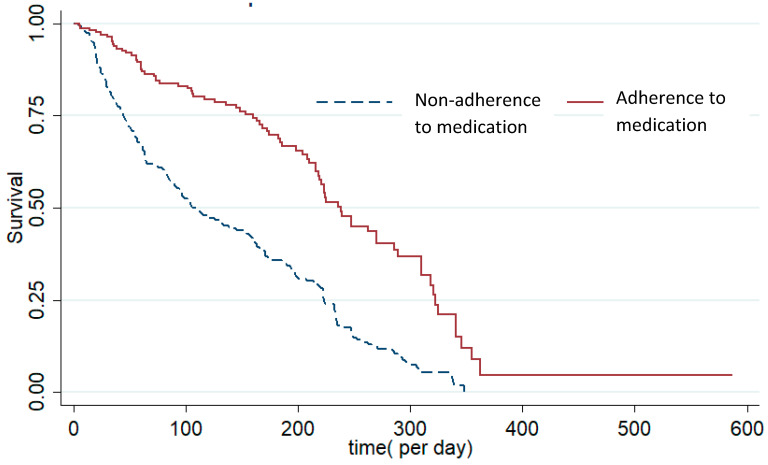
Survival time in days of disease in individuals with schizophrenia spectrum disorder (SSD), separately for medication adherence and medication non-adherence.

**Table 1 ijerph-18-00546-t001:** Distribution of the characteristics, separately by gender.

Variable		Total	Male	Female
N (%)		401 (100)	255 (63.59)	146 (36.40)
		M (SD)	M (SD)	M (SD)
Age		37.81 (9.90)	37.56 (10.06)	38.33 (9.62)
Age at onset of SSD (years)		26.24 (9.05)	25.64 (8.70)	27.29 (9.56)
		Frequency (%)	Frequency (%)	Frequency (%)
Marital status	Married	104 (25.9)	63 (24.7)	41 (28.1)
	Single	250 (62.3)	167 (65.5)	83 (56.8)
	Divorced/widowed	47 (11.7)	25 (9.8)	22 (15.1)
Place of birth	Urban	279 (69.6)	186 (72.9)	93 (63.7)
	Rural	122 (30.4)	69 (27.1)	53 (36.3)
Employment status	Employee	98 (24.4)	92 (36.1)	6 (4.1)
	Unemployed	303 (75.6)	163 (63.9)	140 (95.9)
Substance Abuse	No	306 (76.3)	170 (66.7)	136 (93.2)
	Yes	95 (23.7)	85 (33.3)	10 (6.8)
History of head trauma	No	380 (94.8)	239 (93.7)	141 (96.6)
	Yes	21 (5.2)	16 (6.3)	5 (3.4)
Family history of schizophrenia	No	372 (92.8)	236 (92.5)	136 (93.2)
	Yes	29 (7.2)	19 (7.5)	10 (6.8)
Mode of Onset	Acute	41 (10.2)	28 (11)	13 (8.9)
	Gradual	360 (89.8)	227 (89)	133 (91.1)
Medication Adherence	Non-Adherence	234 (58.4)	149 (58.4)	85 (58.2)
	Adherence	167 (41.6)	106 (41.6)	61 (41.8)
Season of Birth	Spring	120 (29.9)	74 (29)	46 (31.5)
	Summer	123 (30.7)	79 (31)	44 (30.1)
	Autumn	69 (17.2)	47 (18.4)	22 (15.1)
	Winter	89 (22.2)	55 (21.6)	34 (23.3)
History of suicide attempt	No	325 (81.0)	203 (79.6)	122 (83.6)
	Yes	76 (19.0)	52 (20.4)	24 (16.4)
Number of relapses	One to two relapses	222 (55.4)	144 (56.5)	78 (53.4)
	Three to four relapses	114 (28.4)	70 (27.5)	44 (30.1)
	Five to eleven relapses	65 (16.2)	41 (16.1)	24 (16.4)

**Table 2 ijerph-18-00546-t002:** Crude and adjusted Shared Log-Normal Frailty model parameter estimates.

Variable	Categories	Crude HR * (95% CI)	Adjusted HR (95% CI)	*p*-Value
Age of onset of schizophrenia		0.99 (0.98–1.00)	0.99(0.98–1.00)	0.044
Gender	(male) *			
	female	1.00 (0.85–1.18)		
Marital Status	(married)			
	single	1.15 (0.94–1.41)		
	divorced or widowed	1.12 (0.84–1.50)		
Place of Birth	(urban)			
	rural	0.91 (0.75–1.09)		
Employment status	unemployed			
	employed	0.91 (0.74–1.12)		
Substance Abuse	(no)			
	yes	1.08 (0.90–1.31)		
History of head trauma	(no)			
	yes	1.12 (0.79–1.60)		
Family history of schizophrenia	(no)			
	yes	1.23 (0.92–1.65)		
Mode of Onset	(acute)		1	
	gradual	1.48 (1.05–2.08)	1.45 (1.02–2.05)	0.0384
Medication Adherence	(non-adherence)		1	
	adherence	0.68 (0.56–0.83)	0.71(0.58–0.86)	*p* < 0.001
Season of Birth	(spring)			
	summer	1.03 (0.82–1.28)		
	autumn	1.19 (0.94–1.51)		
	winter	1.16 (0.92–1.46)		
History of suicide attempt	(no)		1	
	Yes	1.49 (1.05–2.11)	1.32 (1.09–1.60)	0.0314
Frailty parameter	Sigma Square = 0.08 SE = 0.01 *p* < 0.001			

* HR: Hazard Ratio. In all variables the reference class is in parentheses.

## Data Availability

Data are available upon request to competent experts.

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
