# Peer review of "Identification of Risk Factors to Predict the Occurrences of Relapses in Individuals with Schizophrenia Spectrum Disorder in Iran"

_ijerph, 2021, doi:10.3390/ijerph18020546_

Round 1

Reviewer 1 Report

"Medical chart records of 401 individuals with SSD... A total of 683 relapses were observed in males, and 422 relapses in females."

In addition, for HR's - results are significant only when 95% CI does not include "1".

Author Response

We thank Reviewer #1 for the care devoted to thoroughly review the manuscript. The suggestions and comments helped us to improve the quality of the manuscript. Please find the revised manuscript and the detailed point-by-point-response attached as separate files.

"Medical chart records of 401 individuals with SSD... A total of 683 relapses were observed in males, and 422 relapses in females."

Response: We checked once again the statisctical information, they were all correct; individuals can have multiple relapses.

In addition, for HR's -results are significant only when 95% CI does not include "1"

Response: Thank you! In response, the statistics software rounded up automatically statistical indices to two decimals after the comma. We have corrected this. The text is now: CI : 0.981-0.998

Reviewer 2 Report

This article is about a very relevant topic. Motivation for the current economic, political and spiritual crisis of global peace.

The article is devoted to an extremely relevant topic. The current global world economic, political and spiritual crisis is motivating a rapid increase in diseases associated with mental disorders.

The authors quite rightly point out that social factors play a huge role in the increase in the disease of schizophrenia: a decrease in the standard of living of people and a loss of life prospects. When a person loses his life purpose, his life loses its meaning. In direct proportion to this, the risk of the onset and development of schizophrenia increases.

Author Response

This article is about a very relevant topic. Motivation for the current economic, political and spiritual crisis of global peace.

Response: We thank Reviewer #2 for the positive and encouraging comments.

The article is devoted to an extremely relevant topic. The current global world economic, political and spiritual crisis is motivating a rapid increase in diseases associated with mental disorders

Response: Thank you again for such motivating comments.

The authors quite rightly point out that social factors play a huge role in the increase in the disease of schizophrenia: a decrease in the standard of living of people and a loss of life prospects. When a person loses his life purpose, his life loses its meaning. In direct proportion to this, the risk of the onset and development of schizophrenia increases.

Response: We do fully agree with this point. We note that Reviewer #3 also raised these issues. In response to both Reviewer #2 and #3, we have added information as regards the quality of life of people with SSD. We refer to the text below.

Reviewer 3 Report

I have the following recommendations for the authors to consider and I will review again:

1) Under the Introduction, the authors stated “Despite the progress in describing the brain morphological peculiarities of individuals with SSD, 61 it does not appear that findings from imaging studies were associated with the incidence and 62 prevalence of relapses [3,9,10].” This statement requires scientific fact to support. Please search the following findings under Pubmed to support this claim:
“Between patients with deficit and non-deficit schizophrenia, there were no statistically significant differences in volumetric findings across the various regions of interest.”

2)What is the diagnostic criteria for schizophrenia spectrum disorder? Is it ICD or DSM?

3)The authors should mention about how to increase adherence? They should talk about depot antipsychotics, especially, Long acting injectables (LAI) as compared oral antipsychotics (OA). The authors should search Pubmed for the following findings: adherence values were better in LAI than in OA.

4)The authors need to discuss about the effect of stigma. Please search Pubmed for the following findings and discuss: Asians with mental illnesses were considered as dangerous and aggressive, especially patients suffering from schizophrenia

5)The authors should discuss how to improve quality of life (QoL) of schizophrenia patients. Please search Pubmed for the following findings: interventions to enhance QoL of patients with schizophrenia.

Author Response

We thank Reviewer #3 very much for the care devoted to thoroughly review the manuscript. The comments and suggestions were very helpful to improve the quality of the manuscript.

I have the following recommendations for the authors to consider and I will review again.

Response: Thank you for the valuable and important critiques and suggestions

1)Under the Introduction, the authors stated “Despite the progress in describing the brain morphological peculiarities of individuals with SSD, it does not appear that findings from imaging studies were associated with the incidence and prevalence of relapses [3,9,10].” This statement requires scientific fact to support. Please search the following findings under Pubmed to support this claim “Between patients with deficit and non-deficit schizophrenia, there were no statistically significant differences in volumetric findings across the various regions of interest.”

Response: Thank you. First, please note that the Academic Editor asked for substantial shortening of the Introduction section related to brain imaging studies. In response to Reviewer #3, the following text was added:“....prevalence of relapses (Dazzan et al., 2015; Leucht et al., 2019; Sullivan et al., 2017). On this point, in their exploratory quantitative meta-analysis of structural neuroimaging studies Chee et al. [11]concludedneuroanatomical indices predictive of relapses in individuals with SSS have yet to be identified.

2)What is the diagnostic criteria for schizophrenia spectrum disorder? Is it ICD or DSM?

Response: This was specified as follows:“...ascertained by a trained and experienced psychiatrist or clinical psychologist and based on the DSM 5 (American Psychiatric Association, 2013).2....”

3)The authors should mention about how to increase adherence? They should talk about depot antipsychotics, especially, Long acting injectables (LAI) as compared oral antipsychotics (OA). The authors should search Pubmed for the following findings: adherence values were better in LAI than in OA.

Response: We thank the reviewer for this excellent suggestion and for drawing our attention to the interventions of LAIs. The text reads now:The scientific community proposes two interventions (or their combination)for improving adherence to medication: psychoeducation and long acting injectables (LAIs as opposed to oral antipsychotics (OAPs). With respect to psychoeducation, Bark of et al. [29]....” And: As regards, LAIs, results from systematic reviews and meta-analyses showed that LAIs were superior to OAPs regarding hospitalization rate and all-cause discontinuation of medication but were not superior regarding the risk of hospitalization, time in hospital [35]ornumber of relapses [36]in RCTs, which by definition are less representative of real-world patients in naturalistic studies. Compared to second-generation OAPs, LAIs had lower relapse rates,a longer time to relapse, and fewer hospital days, but also a higher occurrence of extrapyramidal symptoms and prolactin-related symptoms [37]. In this context, 2ndgeneration LAIs had similar effects to 1st-generation LAIs, but the former dramatically increased serum prolactin concentrations, weight and BMI [38]. When compared to placebo, LAIs showed advantages with respect to psychosocial functioning, but this was not the case when compared to OAPs [39]. Compared to OAPs, LAIs reduced the percentage but not the absolute number of hospitalizations [40],while LAIs increased symptoms of dyskinesia [41]. Taken together, previous studies and the present data showed that medication adherence appears to be an important protective factor for relapses, though when using 2nd-generation LAIs in particular the dramatic side-effects such as weight gain, increased prolactin concentrations, and tardive dyskinesia should be carefully considered.

4)The authors need to discuss about the effect of stigma. Please search Pubmed for the following findings and discuss: Asians with mental illnesses were considered as dangerous and aggressive, especially patients suffering from schizophrenia

Response: This is an excellent point; in response, the following text was added:“....and feelings of defeat, social rejection, entrapment, humiliation, and subjectively perceived low social support (O'Connor and Nock, 2014). In this context, and as regards the situation in Iran, individuals with SSD and their caregivers are frequently stigmatized; they can often experience social rejection, covert and open hostility and aggression, and humiliation [52,53]. Given this, individuals with SSD and their caregivers and families risk further stigmatization, social isolation and less access to health care. [54]

5)The authors should discuss how to improve quality of life (QoL) of schizophrenia patients. Please search Pubmed for the following findings: interventions to enhance QoL of patients with schizophrenia

Response: Thank you. In response the following paragraph was added: Fifth, the study did not address an additional but key question from the point-of-view of individuals with SSD and their caregivers: how to improve the quality of life of people with SSD. Besides the strategies for improving treatment adherence discussed above, regular physical activity appears to have a beneficial effect on aspects of SSD and in particular on unpleasant side-effects of APs such as weight gain and sedentary behavior [55-62]. Furthermore, there is a need for early detection and treatment of SSD symptoms so as to keep the quality of life in individuals with SSD as stable as possible [63]

Round 2

Reviewer 1 Report

Dear authors, your manuscript can now be taken into consideration by the editors to be published in the IJERPH.

Author Response

We thank Reviewer #1 for their encouraging statements, and of course we thank for their care devoted to review the manuscript. 
